# Small-Mammal Genomics Highlights Viaducts as Potential Dispersal Conduits for Fragmented Populations

**DOI:** 10.3390/ani14030426

**Published:** 2024-01-28

**Authors:** Tabitha C. Y. Hui, Qian Tang, Elize Y. X. Ng, Ju Lian Chong, Eleanor M. Slade, Frank E. Rheindt

**Affiliations:** 1Asian School of the Environment, Nanyang Technological University, 50 Nanyang Avenue, Singapore 639798, Singapore; 2Department of Biological Sciences, National University of Singapore, 16 Science Drive 4, Singapore 117558, Singapore; dbstq@nus.edu.sg (Q.T.); elizeng@u.nus.edu (E.Y.X.N.); dbsrfe@nus.edu.sg (F.E.R.); 3Discipline of Biological Sciences, School of Natural Sciences, University of Tasmania, Hobart, TAS 7005, Australia; 4Faculty of Science and Marine Environment, Universiti Malaysia Terengganu, Kuala Terengganu 21030, Terengganu, Malaysia; julian@umt.edu.my

**Keywords:** ddRADseq, genetic connectivity, fragmentation, Southeast Asia, local population extinction

## Abstract

**Simple Summary:**

Wildlife crossings are often constructed to enhance genetic connectivity among populations divided by roads (including highways). However, few studies have demonstrated the efficacy of viaducts in counteracting the barrier effects imposed by roads. We measured genetic diversity and divergence in four small mammal species commonly found in rainforests in Malaysia—*Tupaia glis*, *Maxomys rajah*, *M. whiteheadi*, and *Niviventer cremoriventer*—across three treatment types: (1) viaduct sites, at which sampling locations were separated by a highway but connected by a vegetated viaduct; (2) non-viaduct sites, at which sampling locations were separated by a highway and not connected by a viaduct; and (3) control sites, at which there was no road or highway fragmenting the forest. We found that viaducts facilitated movement in small ground-dwelling species such as *M. whiteheadi* and also when existing highways were relatively wide. However, despite the potential for viaducts to facilitate movement and therefore increase genetic connectivity in *M. whiteheadi*, the genetic distance in populations at viaduct sites was still greater than at control and/or non-viaduct sites for the other three species. Our findings highlight the importance of maintaining intact forests rather than relying solely on the construction of viaducts to connect fragmented populations.

**Abstract:**

Wildlife crossings are implemented in many countries to facilitate the dispersal of animals among habitats fragmented by roads. However, the efficacy of different types of habitat corridors remains poorly understood. We used a comprehensive sampling regime in two lowland dipterocarp forest areas in peninsular Malaysia to sample pairs of small mammal individuals in three treatment types: (1) viaduct sites, at which sampling locations were separated by a highway but connected by a vegetated viaduct; (2) non-viaduct sites, at which sampling locations were separated by a highway and not connected by a viaduct; and (3) control sites, at which there was no highway fragmenting the forest. For four small mammal species, the common tree shrew *Tupaia glis*, Rajah’s spiny rat *Maxomys rajah*, Whitehead’s spiny rat *Maxomys whiteheadi* and dark-tailed tree rat *Niviventer cremoriventer*, we used genome-wide markers to assess genetic diversity, gene flow and genetic structure. The differences in genetic distance across sampling settings among the four species indicate that they respond differently to the presence of highways and viaducts. Viaducts connecting forests separated by highways appear to maintain higher population connectivity than forest fragments without viaducts, at least in *M. whiteheadi*, but apparently not in the other species.

## 1. Introduction

Habitat fragmentation, degradation, and loss pose the most significant threats to the structure and persistence of animal populations and communities [1,2]. Fragmentation is most rapid in developing countries where the expansion of road networks is increasing due to competing land uses such as farming, manufacturing and housing [3]. Barriers that bisect continuous habitat, particularly roads (including highways), initiate the process of habitat fragmentation and can restrict or eliminate animal movement through a landscape [4], with concomitant consequences for connectivity and gene flow [5]. The reduced connectivity may lead to a decrease in viability and persistence of isolated populations [6]. However, the extent to which habitat fragmentation has a negative effect on the genetic structure and persistence of animal populations remains debated. While some studies have demonstrated a negative impact of habitat fragmentation on species’ population genetic structure [5,7], others have failed to detect these effects [8].

Wildlife corridors are widely understood to connect habitat fragments and mediate the effects of fragmentation [9]. They include naturally occurring linear habitats such as riparian reserves and hedgerows, purpose-built structures such as wildlife overpasses and underpasses, and incidental structures such as drainage culverts. Corridors have been widely advocated as essential components of reserve design because they can connect isolated areas of suitable habitat and thus minimise the harmful effects of habitat fragmentation on animal movement [10,11,12,13]. Corridors are predicted to benefit populations in patchy habitats by promoting movement, which increases population densities, gene flow, and recolonisation of extinct patch populations [14]. However, the efficacy of such passages remains largely untested [15,16,17], and there has been much debate about their effectiveness in connecting isolated populations [18,19]. Corridors have been shown to increase connectivity, maintain biodiversity [20], increase population sizes [21,22,23], facilitate movement between fragmented patches [10,18,20,24,25,26,27,28,29,30] and promote gene flow [28,31,32]. Other studies, however, have found no significant effects of corridors or even negative impacts [33,34,35].

To demonstrate the effectiveness of wildlife corridors in increasing connectivity between populations, studies of genetic relatedness are recommended. Molecular techniques such as next-generation sequencing, which has recently been used to reveal fine-scale population structure [36,37], offer new promise in investigating the influence of fragmentation and barriers on population connectivity [38,39]. Genetic methods allow us to measure average migration rates over time, which reveals the effects of fragmentation over several generations and is not as sensitive to current population sizes as mark-recapture studies are (e.g., when populations are extremely low, mark-recapture studies may be impossible) [40]. In addition, molecular techniques measure effective dispersal, the amount of gene flow between populations [41]. As genetic techniques use a single temporal sample per population to estimate migration rather than multiple samples, these techniques require less field effort than mark-recapture [28].

The consequences of habitat fragmentation on dispersal and genetic diversity are still largely unknown for non-volant small mammals, especially in some of the world’s equatorial rainforest areas, such as peninsular Malaysia in Southeast Asia. The relative scarcity of data on the effects of fragmentation on small mammals in Southeast Asia is alarming, as these animals provide important ecosystem functions and services for their natural habitats [42]. They are important seed dispersers, pollinators, invertebrate and seed predators, as well as prey for larger predators. Small mammal communities provide a good model for studying such impacts because species in these communities generally use a wide variety of resources, have short generation times that allow for quick detection of environmental change, may be permanent residents of a site, and usually respond to disturbances in a perceptible and measurable way [43]. Smaller mammals are thought to be particularly susceptible to fragmentation due to their limited ability to travel over long distances through exposed habitats [44]. They are thus likely to suffer severe impacts of fragmentation and would benefit from any increases in connectivity brought about by the construction of wildlife corridors.

In this study, we assessed the constraints in migration and gene flow due to habitat fragmentation caused by the construction of highways, and the effectiveness of wildlife underpasses, known in Malaysia as eco-viaducts, in facilitating movement and genetic connectivity in small mammals by comparing genetic distances (a measure of genetic differences, computed by using allele frequency data from many different loci) between individuals at viaduct sites, non-viaduct sites and control sites. We assumed that genetic relationships at all sites would have been at similar levels of divergence prior to fragmentation. We hypothesized that: (1) for forest species, the genetic distance between individuals should be lowest for populations in intact forest and highest for populations fragmented by highways; (2) if eco-viaducts are effective in maintaining population connectivity, the genetic distance between individuals connected by an eco-viaduct will more closely resemble the distance between individuals in intact forest; (3) alternatively, if eco-viaducts are not effective, then pairwise genetic distances should be similar to those in pairs separated by highways. By examining the population genetic structure of small mammal species across different spatial settings in this study, we aim to determine the effectiveness of eco-viaducts across highways in maintaining genetic linkage in a fragmented landscape.

## 2. Materials and Methods

### 2.1. Study Sites

We conducted this study in Kenyir, Terengganu (Figure 1a, elevation 100–300 m) and Sungai Yu, Pahang (Figure 1b, elevation 130–210 m), Peninsular Malaysia. Terrain is hilly and consists mostly of lowland dipterocarp forest. Rainfall averages 3000 mm per year with a pronounced wet season from November to March. Flooding is common during this period. The study areas are gazetted as forest reserves and can be logged under permit. Kenyir and Sungai Yu adjoin the Taman Negara National Park, Malaysia’s first national park, to the north and west, respectively (Figure 1c). Kenyir and Sungai Yu have rich biodiversity, but are prone to illegal logging, conversion to plantations, and poaching as they do not have the same protection status as a national park. Kenyir forest is bisected by federal route 185 (henceforth highway 185) (Figure 1a) while Sungai Yu forest is bisected by federal route 34 (henceforth highway 34) (Figure 1b). Highway 185 is a two-lane single carriageway (one lane in each direction) with a width of about 8 m and shoulder width of 2 m on either side. Highway 34 is a four-lane dual carriageway (two lanes in each direction) with a width of about 20 m and shoulder width of about 1–2 m on either side. There was no fencing along these highways, although vehicle guard rails were present in certain sections. The speed limit was 90 km/h on highway 185 and 110 km/h on highway 34. Average (±s.e.) vehicular traffic (counted at two different points on each highway between 0700 and 2300 h on six separate days) was 23 ± 5 cars, 16 ± 5 motorcycles and 10 ± 4 heavy vehicles per hour on highway 185; and 52 ± 11 cars, 24 ± 8 motorcycles and 20 ± 6 heavy vehicles per hour on highway 34. Many wildlife crossings, primarily eco-viaducts, have been constructed across Malaysia to restore connectivity between highway-bisected forest fragments, including forests in Kenyir and Sungai Yu. Eco-viaducts are bridge-like elevated roads (and highways) that allow passageway beneath for wildlife to safely cross between forests on either side of the highway (Figure 1d,e). Three eco-viaducts have been constructed across highway 185 in Kenyir and another three across highway 34 in Sungai Yu. The Kenyir eco-viaducts measuring 245 m, 140 m and 245 m in length were completed in 2008 (Figure 1a), while the Sungai Yu eco-viaducts measuring 80 m, 300 m and 1000 m were completed in 2014 (Figure 1b).

### 2.2. Small Mammal Trapping

Eighteen sites, six of them being non-viaduct highway-side sites, another six viaduct sites and six control sites (>500 m from any roads and highways), were selected for this study (Figure 1a,b). Both Kenyir and Sungai Yu had a set of nine sites each (Figure 1a,b). Each site was characterized by a pair of grids with 20 traps, each grid within a pair on opposite sides of the highway. The 20 traps within a grid were composed of ten Elliott sheet metal traps (32 × 10 × 10 cm) and ten Tomahawk wire cage traps (48 × 15 × 15 cm) (Appendix A) (except the 80 m wide viaduct in Sungai Yu which had only ten traps as there was not enough width across the viaduct to place 20 traps 10 m apart). Alternating Elliott and cage traps were set in parallel grids, with 10 m between traps (Figure 2). Paired grids were 50 m apart from each other, separated by the highway. The distance between grids was roughly chosen to be identical to the distance required for a crossing of the highway and its verges and ditches to confirm whether the highway accounts for any inhibition of movement. To avoid pseudoreplication issues, pairs of grids were at least 500 m apart, which is more than the home range of the small mammals in our study [45]. Trapping was conducted four times at Kenyir in 2017 and four times at Sungai Yu in 2018 from March to November, during the drier inter-monsoon and southwest monsoon seasons. Traps were set, checked and rebaited with bananas and peanut butter (cage traps) and vanilla scented oats (Elliott traps) [46] during five consecutive mornings and evenings to assess the diversity and abundance of both nocturnal and diurnal mammals. Captured individuals were identified to species, sex, age class and reproductive condition, weighed, measured and ear tagged before being released at the trap site. We collected ear clips for DNA tissue sampling from all animals trapped and tagged during live trapping using a 2 mm ear punch. Tissue samples were stored in absolute ethanol.

In total, we trapped 448 individuals from 17 species. The species Whitehead’s spiny rat *Maxomys whiteheadi* (Figure 3a), Rajah’s spiny rat *Maxomys rajah* (Figure 3b), dark-tailed tree rat *Niviventer cremoriventer* (Figure 3c) and common tree shrew *Tupaia glis* (Figure 3d) were included in this study to capture a wide variety of biological and ecological characteristics. These four small mammal species are found in tall lowland forests and forest edge, and feed on insects and plant matter such as fruits and seeds. *M. rajah* and *T. glis* are similar in size and weight (100–200 g), while *M. whiteheadi* (35–80 g) and *N. cremoriventer* (50–100 g) are smaller. *M. whiteheadi* and *M. rajah* are both nocturnal ground dwelling rats; *N. cremoriventer* is a nocturnal tree rat which is a good climber and lives both arboreally and on the ground, *T. glis* is a diurnal treeshrew active on the ground and in the understory. Samples from 32 individuals (7 in Kenyir, 25 in Sungai Yu) of *M. rajah*, 39 (16 in Kenyir, 23 in Sungai Yu) of *M. whiteheadi*, 22 (20 in Kenyir, 2 in Sungai Yu) of *N. cremoriventer* and 45 (25 in Kenyir, 20 in Sungai Yu) of *T. glis* were selected based on their occurrences in pairs of sites (Appendix A).

### 2.3. DNA Extraction and ddRAD-Seq Library Preparation

DNA extractions were performed using the DNeasy Blood & Tissue Kit (QIAGEN, Hilden, Germany) following the manufacturer’s protocol for tissue. We prepared two libraries following Ng et al.’s [47] double-digest restriction-associated DNA sequencing (ddRADseq) protocol using EcoRI and MspI. To select for 250–600 bp fragments, as well as for the clean-up steps, we used Sera-Mag magnetic beads (Thermo Scientific, Waltham, MA, USA). DNA quantifications were performed with a Qubit 2.0 High Sensitivity DNA Assay (Invitrogen, Waltham, MA, USA). Before pooling samples, we checked DNA fragment size using a Fragment Analyser (Advanced Analytical Technologies, Inc., Ames, IA, USA). The two libraries were then spiked with 5% PhiX to prevent low nucleotide diversity issues from affecting the quality of the data, and were subsequently sequenced on an Illumina HiSeq 4000 platform (150 bp paired-end run).

Reads were demultiplexed and trimmed to 145 bp with *process_radtags* in STACKS 1.42 [48]. Reads with one or more uncalled bases were removed. For reference-based identification of ddRAD loci, we first aligned the demultiplexed reads of *T. glis* to the closely related *Tupaia chinensis* (GCA_000334495.1) [49], using the Burrows-Wheeler Aligner (BWA) [50] to index this reference genome. We used samtools 1.3.1 to convert the *sam* files to *bam* files, sort the aligned reads according to coordinates and filter files with a minimum required mapping quality score (MAPQ) score of 20 [51]. To call and filter single nucleotide polymorphisms (SNPs), we used *ref_map.pl* and *population* in STACKS 1.42 [48] for *T. glis*. In *population*, we set stack depth to 10 and the percentage of individuals represented at each locus to 0.9 and admitted only one random SNP per locus to preclude analysis of linked SNPs. For the other three species, there was no suitable reference genome, so SNPs were called de novo. In *ustacks*, we set the maximum distance (in nucleotides) allowed between stacks to 2, minimum depth of coverage required to create a stack to 3 and maximum distance allowed to align secondary reads to primary stacks to 4. We checked for SNPs under selection using BayeScan 2.1 [52] and used Plink 1.90 to remove linked loci and to calculate the level of missing data [53]. We allowed a variety of filters (0% or 10% missingness, including or excluding linked loci of *r*^2^ > 0.5, and including or excluding minor allele frequency < 5%) to generate eight datasets for the preliminary testing to rule out potential sampling artefacts (e.g., non-random distribution of genotypes) (Appendix A).

### 2.4. Statistical Analyses

To visualise genetic differentiation amongst individuals and identify potential population subdivision, we performed principal component analysis (PCA) using the R package SNPRelate 1.6.6 [54] for all eight datasets (four species at two study areas). We carried out sensitivity analysis, checking across various settings of missingness, linkage and minor allele frequencies by confirming the consistency of PCAs across eight datasets for all four species. As a consequence of this sensitivity analysis, we selected the dataset with 10% missingness, in which we excluded linked loci but included minor alleles for all subsequent analyses (Appendix A). To understand the genetic differentiation between the two study areas, we calculated the Weir-Cockerham’s *F_ST_* between the two study areas using VCFtools v4.1 [55] and individual-pairwise relatedness using maximum likelihood estimation as implemented in SNPRelate.

We calculated pairwise genetic distances between all individuals within a species with the R package *poppr* [56]. To evaluate the efficacy of viaducts in facilitating the dispersal of small mammals and test the barrier effects of highways, Kruskal–Wallis tests were conducted on the genetic distances of individuals between pairs of grids within a site for each species across the three treatment types (viaduct, non-viaduct and control) and two study areas. This means that every individual of one species on one side of the highway or pair (in the case of control sites) is compared with every other individual of the same species on the other side of the highway or pair in the same pair of grids (site).

We also compared pairwise genetic distances among all individuals within a species in relation to different spatial distances to determine how resistance to dispersal changes in different landscape types. Using the Least Cost Path function in ArcGIS 10.6.1, we modeled three types of spatial distances, namely Euclidean distance (for examining the effects of isolation by distance (IBD)), least cost distance considering roads (and highways) as agents of resistance (not considering the potential effects of viaducts), and least cost distance considering roads as agents of resistance and with viaducts facilitating movement. The resistance value for non-road areas and viaducts was set at 0 to represent low-cost distance, and for roads and water bodies set at 1 to represent the high cost of crossing roads and water bodies in the least-cost distance calculations. We calculated the correlations between genetic and spatial distances for each species at each treatment grid using Mantel tests with 99 permutations (default value) and Monte-Carlo corrected *p* values, using the R package *ade4* [57]. One-tailed tests were used as the IBD model predicts that genetic differentiation will be positively correlated with increasing geographic distance [58]. All statistical analyses were conducted in R version 4.0.2 (R Core Team, 2020). Significance was considered at the α = 0.05 level.

## 3. Results

We obtained ~9000 SNPs using STACKS for the complete data set consisting of all four species and retained ~5000 SNPs, after removing SNPs under selection and disequilibrium (Appendix A). For each species, PCAs across different datasets were generally consistent, therefore, we chose the SNP dataset filtering 10% missingness and absolute linkage (>0.95) for all subsequent analyses.

PCA did not reveal any clear patterns of clustering in genetic variation amongst viaduct, non-viaduct and control sites (Figure 4, Appendix A). However, PCA revealed slight genomic differentiation in *M. whiteheadi* between study areas along principal component 1 (Figure 5). All four species exhibited low genetic differentiation (*F_ST_*) between the two study areas (*M. whiteheadi* F_ST_ = 0.022113378, *M. rajah* F_ST_ = 0.055480442, *N. cremoriventer* F_ST_ = 0.069197229, *T. glis* F_ST_ = 0.015547675). We found related individuals (relatedness > 0.1) between the two study areas in *M. rajah* and *T. glis*, suggesting occasional exchange of individuals between the two study areas.

Kruskal–Wallis tests revealed that there were differences in the ranking of treatment types by genetic distance for *M. whiteheadi* (*χ*^2^_2,79_ = 28.22, *p* < 0.0001, Figure 6a) and *M. rajah* (*χ*^2^_2,87_ = 39.01, *p* < 0.0001, Figure 6b). Genetic distances between individuals were lowest at viaduct sites for *M. whiteheadi*. For *M. rajah*, genetic distances were highest at viaduct sites and lowest at non-viaduct sites, with control sites in between. There were no significant differences in genetic distances between treatment types for *N. cremoriventer* (*χ*^2^_2,27_ = 5.38, *p* = 0.068, Figure 6c) and *T. glis* (*χ*^2^_2,101_ = 3.39, *p* = 0.18, Figure 6d).

Study area was also a significant factor influencing genetic distances between individuals within a site for *M. whiteheadi* (*χ*^2^_1,79_ = 7.18, *p* < 0.01, Figure 7a), with individuals from Sungai Yu showing more genetic differentiation than at Kenyir. Likewise, *T. glis* also showed significantly more genetic differentiation at Sungai Yu (*χ*^2^_1,101_ = 37.35, *p* < 0.0001, Figure 7d). *M. rajah* (*χ*^2^_1,87_ = 0.72, *p* = 0.40, Figure 7b) did not show differences in genetic distances between study areas. Due to low sample sizes at Sungai Yu, only *N. cremoriventer* samples from Kenyir were included in the analysis (Figure 7c).

The dispersal of *M. whiteheadi* was found to be most consistent with isolation by distance (IBD) (highest correlation with Euclidean distance), indicating genetic and spatial distances between individuals were significantly positively correlated in both Kenyir and Sungai Yu, with minimal barrier effects of roads or facilitation effects of viaducts (Table 1). In Sungai Yu, the dispersal of *M. rajah* and *T. glis* correlated best with least cost distance, considering roads as resistance and viaducts as corridors facilitating movement, showing that they apparently prefer using viaducts to cross road barriers. No significant correlation with spatial distances were found in Kenyir for *M. rajah*, *T. glis* and *N. cremoriventer*. The paired samples of *N. cremoriventer* in Sungai Yu were insufficient for comparisons (Table 1).

## 4. Discussion

Our study showed that eco-viaducts may facilitate movement and the maintenance of gene flow in a landscape bisected by roads at least in some species such as *M. whiteheadi*. Presumably, the higher genetic similarity of individuals at viaduct sites in these species, relative to that at non-viaduct sites, is due to movement of individuals through the viaducts [59], thus reducing population subdivision caused by habitat fragmentation.

Highways did not seem to pose a barrier to gene flow in *M. rajah* (Figure 6b). Surprisingly, the greatest genetic differentiation amongst the three treatment types for *M. rajah* was at viaduct sites. All spatial distance model correlations were significant as well, indicating that viaducts were not essential for *M. rajah* to cross roads. This differential impact of roads is likely because for a less agile ground dwelling small mammal such as *M. rajah*, the effort taken to cross natural structures in forests such as dense vegetation and uneven ground, or a longer route to reach a viaduct and go through it, could be similar to crossing artificial structures such as roads (Figure 6b). Perhaps *M. rajah* only uses the viaducts for dispersal to other more favourable habitats, as viaducts may also facilitate the movements of its predators such as civets and leopard cats [60].

Movement and gene flow in *M. whiteheadi* were influenced by road barriers as well as the study area. The *M. whiteheadi* population in Kenyir was slightly differentiated from the population in Sungai Yu. Their dispersal was also most aligned with the Euclidean distance model in both study areas, pointing to IBD. A distinction between populations was not observed in the other three species. A lack of differentiation between the two study areas for the other three species is not surprising as the study areas are relatively near, there are no severe breaks in forest cover at a landscape scale, and genetic differentiation is not only the result of isolation by distance. This is why we chose to use individual-based spatial analyses to investigate the subtle differentiation.

Greater genetic connectivity was observed for both *M. whiteheadi* and *T. glis* at Kenyir than at Sungai Yu. The highway at Sungai Yu is more than double the width of the highway at Kenyir and it is probably more of a barrier to movement than the narrower Kenyir highway. Thus, the significance of viaducts in facilitating movement and increasing genetic connectivity across wider roads such as at Sungai Yu would be greater than at Kenyir. This was probably the case for *M. whiteheadi* as it also showed the lowest genetic differentiation in viaduct sites. Captures of *M. whiteheadi* individuals within the viaducts themselves suggest that *M. whiteheadi* does use viaducts to cross roads and may even have part of their home ranges overlapping the viaducts. Of interest was the observation that for *T. glis*, individuals at viaduct sites were still genetically more dissimilar than those at control sites. This shows that despite the addition of viaducts to connect fragmented populations, they will not be able to restore genetic connectivity to match that of the original intact forest.

In Kenyir, the dispersal of *M. rajah*, *N. cremoriventer* and *T. glis* did not correlate significantly with any of the least cost distance models (Table 1), suggesting that the dispersal ability of these three species goes beyond the spatial scale of the study in this area. The highway bisecting the forest in Kenyir is much narrower than the highway in Sungai Yu, likely presenting itself as less of a barrier to movement. *M. rajah* has been found as road kills in Kenyir, evidence that it does cross roads directly if narrow enough [59]. *T. glis* is a relatively mobile species, able to disperse >4 km [61]; one individual in this study was trapped in two viaducts, a distance separation of >5 km.

There was no clear pattern of clustering in genetic variation amongst viaduct, non-viaduct and control sites in intact forest (Figure 4, Appendix A), suggesting that there are no significant differences in genetic composition of small mammal species amongst the three habitats. However, there was slight genetic differentiation between the *M. whiteheadi* populations at Kenyir and Sungai Yu (Figure 5). As the smallest species in this study, it is expected to have the least dispersal ability and may be starting to show genetic divergence between populations separated by greater distances.

The upgrading of the highways and construction of the viaducts in this study were relatively recent; therefore, population genetic data may not fully reflect the evolutionary change if the generation time is long. It is possible that not enough time has passed to allow the vegetation in the viaducts to grow and establish stable corridors or habitats. Mills and Allendorf [62] suggested that only one migrant per generation is sufficient to maintain genetic diversity while allowing some divergence between populations. Rosenberg et al. [24] also argue that corridors are more effective at maintaining movement between populations for habitat specialists rather than for habitat generalists. All four species in this study are tolerant to disturbed forests and forest edge, and their diets are varied, consisting of insects, fruit and seeds. This may explain why the viaducts did not increase genetic connectivity as much as expected.

Our study shows that viaducts can increase genetic connectivity, but are not effective across all species and roads or highways. The effectiveness of the viaducts in reducing the barrier effects of roads depends on the target species’ dispersal abilities, generation time, perception of landscape characteristics, road width, viaduct age and vegetation structural maturity. Viaducts work better when they have a structure which is more conducive for movement than a direct crossing across the road. However, from this study we observed that viaducts were not able to restore genetic connectivity to match that of the original intact forest and should therefore only be used as a last resort if it is not possible to keep the forest intact.

## 5. Conclusions

Our study shows that wildlife crossings such as viaducts may assist in maintaining gene flow within populations of certain small mammals in areas with relatively wider roads. In a managed landscape, there must be a balance between the economic benefits of building roads and the ecological benefits to species in maintaining connectivity. We have shown that one of the benefits of constructing and maintaining viaducts is an increase in gene flow within populations in some species, which may result in an increase in population persistence and a decrease in inbreeding.

## Figures and Tables

**Figure 1 animals-14-00426-f001:**
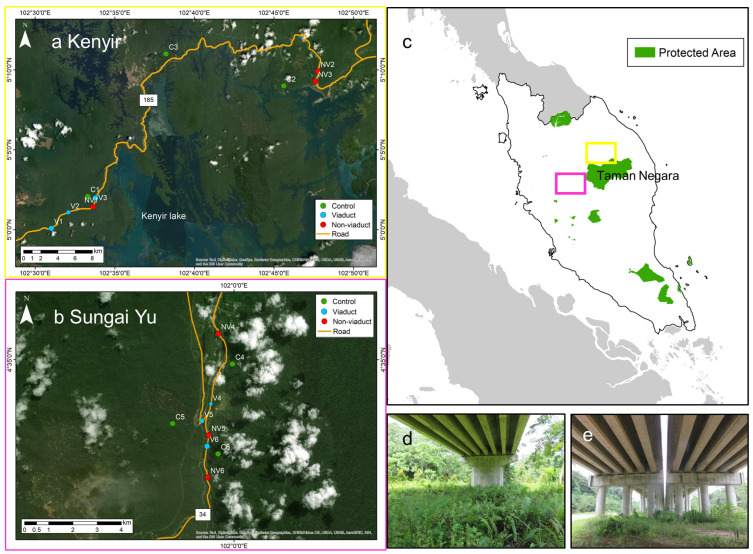
Sampling sites in (**a**) Kenyir, (**b**) Sungai Yu. Sites situated away from highways, along highways and adjacent to eco-viaducts are named “Control”, “Non-viaduct” and “Viaduct”, respectively. (**c**) Locations of study areas in Peninsular Malaysia (yellow: Kenyir; pink: Sungai Yu). Pictures of eco-viaducts (wildlife crossings) at (**d**) Kenyir and (**e**) Sungai Yu (photos by Tabitha Hui).

**Figure 2 animals-14-00426-f002:**
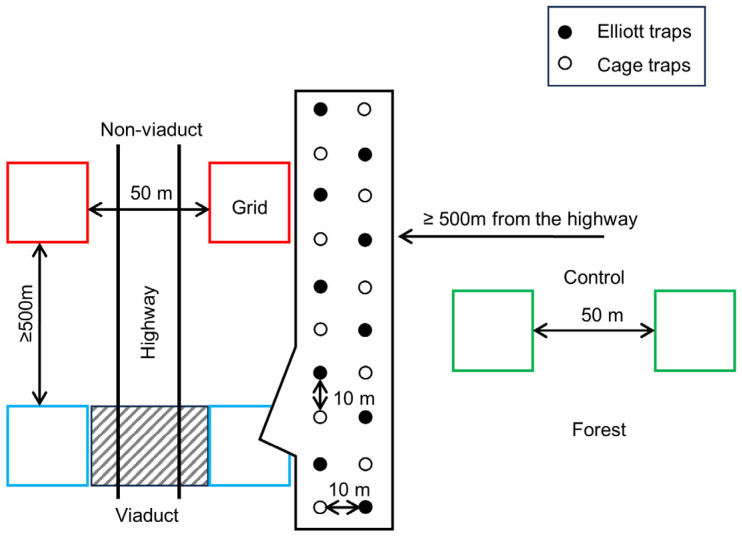
Schematic diagram of sampling design. The expanded grid box shows the layout of traps. Blue boxes: viaduct sites, red boxes: non-viaduct sites, green boxes: control sites.

**Figure 3 animals-14-00426-f003:**
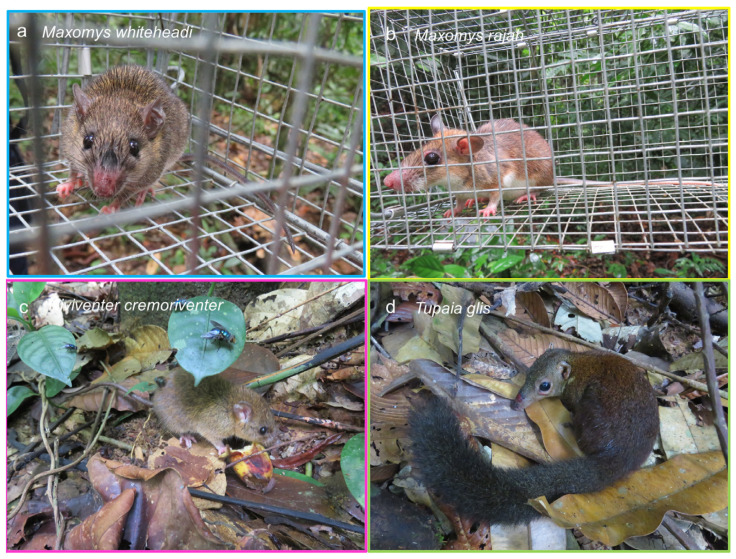
The four study species: (**a**) *Maxomys whiteheadi*, (**b**) *Maxomys rajah*, (**c**) *Niviventer cremoriventer* and (**d**) *Tupaia glis*. Photos by Tabitha Hui.

**Figure 4 animals-14-00426-f004:**
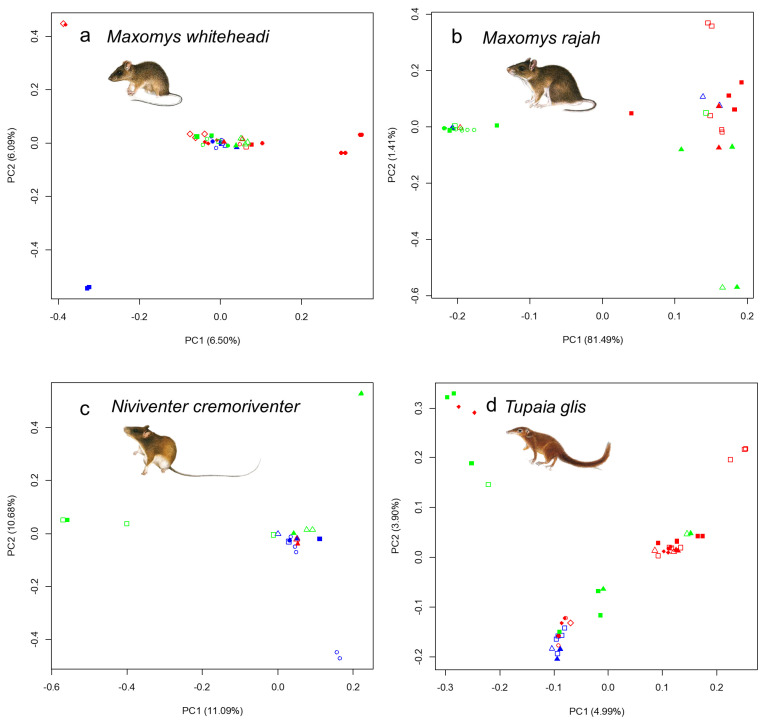
Principal component analysis (axes PC1 and PC2) of genetic differentiation among sampled individuals. (**a**) *Maxomys whiteheadi*, (**b**) *Maxomys rajah*, (**c**) *Niviventer cremoriventer* and (**d**) *Tupaia glis*. Blue: viaduct sites, red: non-viaduct sites, green: control sites. Same shape: same pair, filled/open shapes: opposite sides of the road of the same pair.

**Figure 5 animals-14-00426-f005:**
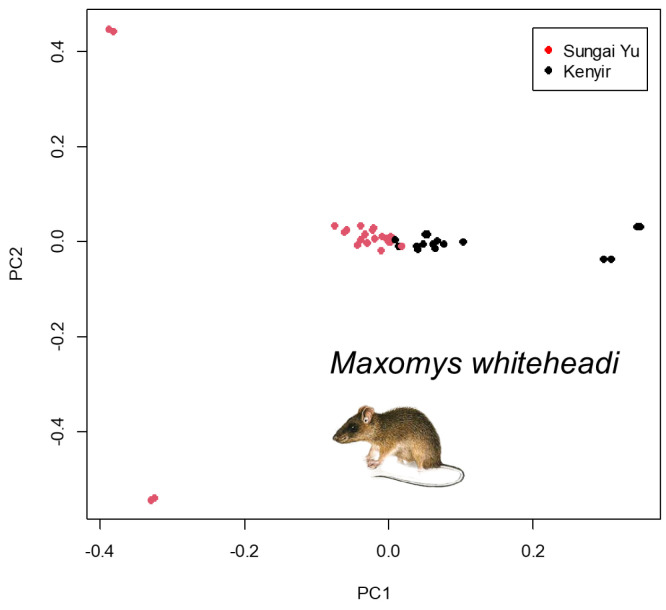
Principal component analysis (axes PC1 and PC2) of genetic differentiation among sampled individuals of *Maxomys whiteheadi* showing slight separation between Sungai Yu and Kenyir populations.

**Figure 6 animals-14-00426-f006:**
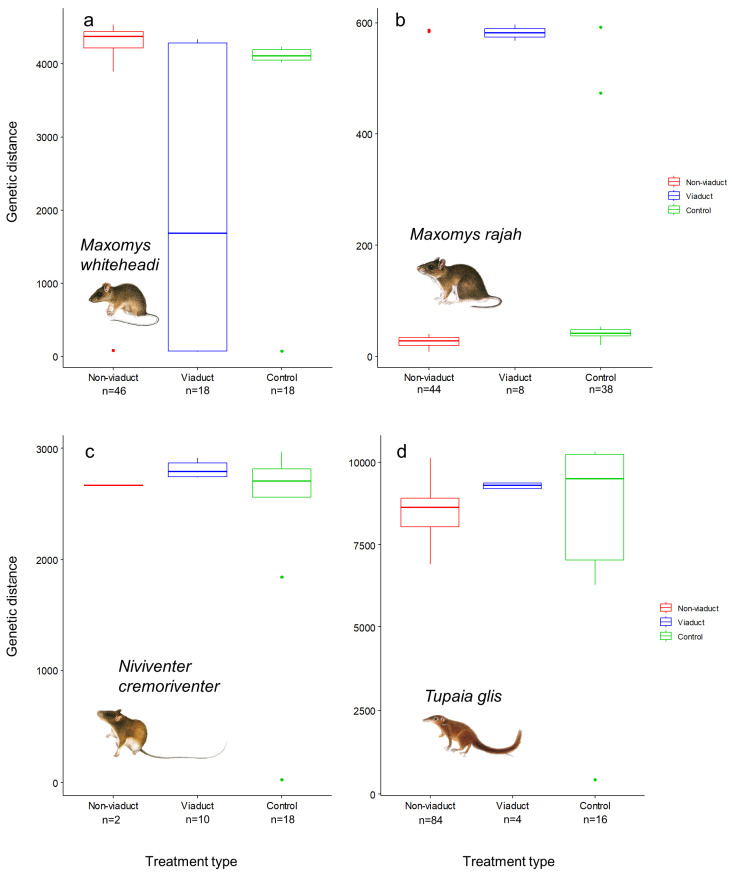
Comparisons of genetic distances between non-viaduct, viaduct and control sites. (**a**) *Maxomys whiteheadi*, (**b**) *Maxomys rajah*, (**c**) *Niviventer cremoriventer* and (**d**) *Tupaia glis*. The boxes, lines in the middle, whiskers and dots represent the interquartile range, median, minimum and maximum values (no more than 1.5 times the interquartile range) and outliers, respectively.

**Figure 7 animals-14-00426-f007:**
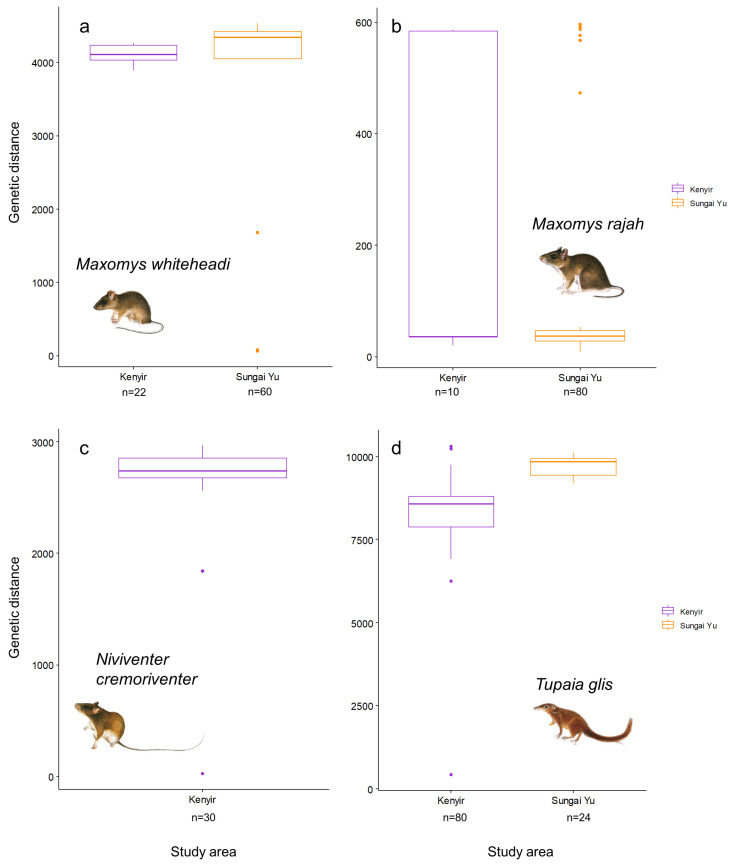
Comparisons of genetic distances between study areas. (**a**) *Maxomys whiteheadi*, (**b**) *Maxomys rajah*, (**c**) *Niviventer cremoriventer* and (**d**) *Tupaia glis*. The boxes, lines in the middle, whiskers and dots represent the interquartile range, median, minimum and maximum values (no more than 1.5 times the interquartile range) and outliers, respectively.

**Table 1 animals-14-00426-t001:** Mantel’s r correlations between the genetic distances of each species and each spatial distance model (Euclidean: Euclidean distance; Road: least cost distance considering roads; Viaduct: least cost distance considering roads as resistance and viaducts facilitating movement). Results in bold are the spatial distance models with the highest correlation for the species in each study area, where multiple spatial distance models were significant.

		Kenyir		Sungai Yu	
	Spatial Distance Model	Mantel *r*	Simulated *p*-Value	Mantel *r*	Simulated *p*-Value
	Euclidean	0.118	0.15	0.355	0.01
*Maxomys* *rajah*	Road	0.117	0.16	0.373	0.01
	Viaduct	0.117	0.14	**0.506**	0.01
	Euclidean	**0.245**	0.01	0.082	0.02
*Maxomys whiteheadi*	Road	0.237	0.01	0.124	0.19
	Viaduct	0.237	0.01	0.126	0.22
	Euclidean	0.096	0.1		
*Niviventer cremoriventer*	Road	0.112	0.12		
	Viaduct	0.117	0.1		
	Euclidean	0.121	0.17	0.113	0.17
*Tupaia glis*	Road	0.102	0.19	0.204	0.09
	Viaduct	0.101	0.23	**0.207**	0.05

## Data Availability

All available data are presented in this manuscript.

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
