# Peer review of "Small-Mammal Genomics Highlights Viaducts as Potential Dispersal Conduits for Fragmented Populations"

_animals, 2024, doi:10.3390/ani14030426_

Round 1

Reviewer 1 Report (New Reviewer)

Comments and Suggestions for Authors

This paper tests the importance of viaducts for wildlife connectivity for four common small mammal species in Malaesya. 

Title: Landscape genomics - isn't small mammal genomics more accurate since genetic distance and divergence has been studied for small mammals? I recommend rewording the title

Simple Summary: 

It is unclear whether or not for the species M. whiteheadi the genetic distance between the two populations separated by road but connected by viaduct is significant.

Abstract: is much clearer than simple summary but repeats it to a very large extent. I recommend to drop Simple Summary.

Keywords: conservation planning; protected area; local population extinction - are surprising since there is no mention of them in the abstract, in the introduction or in the discussion, I recommend their exclusion.

Introduction:

It is well written clear and sufficiently well documented. However, it is not clear whether the study considers different types of roads (the phrase roads appears frequently) or whether the study only refers to motorways. I recommend consistency in the use of syntagms. 

Materials and Methods

Study sites:

The description of roads is quite ambiguous in that different values of road width appear it is not clear whether they have more than one lane in one direction of travel or there are more than one lane in each direction of travel. Also some information is needed on how they are protected, whether they have protective fencing or not, whether they are made of wire netting or not, and how large the mesh size is especially for the species M. whiteheadi (35-80 g) and N. cremoriventer (50-100 g) is a nocturnal tree rat which is a good climber and lives both arboreally and on the ground. In particular it is important to note whether they are indeed barriers to small mammals or not. Many small mammal species pass through wire mesh and often manage to cross highways. Also some information on the flow of cars on motorways would give a more complete picture of the ability of small mammals to cross motorways.

Figure 1c - blue and red colours are missing in the figure

It would be clearer if the legend in fig 1c instead of protected area said national park

Small mammal trapping: the sampling protocol is clear. I suggest adding some photos in the supplementary material with the types of traps used. 

Images of the four species that are included in the study are welcome but a mention of the source of the images (original or not) is needed.

Results:

For Table 1 the rendering of p-values and use of red colour for significant results is redundant. 

The only major uncertainty about the results, which creates a suspicion as to the relevance of the results, is whether or not in the two areas studied roads are indeed a barrier for the four species considered. Whether there were significant differences between the genetic structure of the control populations and the viaduct or road populations for the species considered. And if they do not constitute barriers then we cannot test the importance of viaducts for these species.

Dicutations are clearly written and are supported by the results. A discussion of the difference between the two areas in Maxomys whiteheadi, Figure 5 is expected.

Overall this is an interesting paper with important practical implications for biodiversity conservation and ensuring gene exchange between isolated populations. But at this stage I suggest first testing whether roads are really barriers for these species and then testing the importance of viaducts. 

Author Response

Reviewer 2 Report (New Reviewer)

Comments and Suggestions for Authors

Landscape genomics highlights viaducts as potential dispersal conduits for fragmented small-mammal populations

Tabitha C.Y. Hui et al.

Review

In the study authors tested constraints of small mammal migration imposed by highways and effectiveness of eco-viaducts in maintaining gene flow across the roads. They selected viaduct containing sites, non -viaduct sites and control sites in the tropical forests of Malaysia. Results differed between species, therefore giving good thoughts to discuss fragmentation-related issues.

It is a nice paper to be published in Animals. I appreciate methods and data treatment, though, some outcomes might be related to insufficient sample sizes of analyzed species, especially of M. whiteheadi in Sungai Yu and M. rajah in Kenyir. Sample size was mentioned, however, I propose to add more text to emphasize possibility of differences under larger sampling amounts.

Please found comments below, requiring minor revision of the paper before acception.

General comments

Only one of the mentioned files was attached as supplement. The other supplemental files (tables and figures) are missing on the site.

Please supply explanation for meaning of box, whiskers and dots in the legends of corresponding figures.

Simple summary

Line 26: delete “and”

Line 29: and for the fourth species? Genetic distance for Maxomys whiteheadi was not mentioned in Lines 25, 26.

Abstract

Please present Latin names in full

Introduction

Line 54: [1,2] – check template and change throughout

Line 73: [10–13], change throughout

Line 104: small mammal, no hyphen between words

Material and methods

Line 136: use en dash

Line 196: common and Latin names of all species should be given on the first use

Results

Line 316: explain, why Kruskall-Wallis test yields chi-square, not H?

Discussion

Line 378: there can be no “slightly” difference; it is significant or not significant

Lines 405 to 407 – this was explained above

Conclusions

Lines 444 and 445: I propose text in yellow need to be deleted.

Back matter

Only table S1 is present in Supplementary materials

Data availability statement is missing, delete if not needed

References

Are not formatted according requirements of the journal

Round 2

Reviewer 1 Report (New Reviewer)

Comments and Suggestions for Authors

Consider the manuscript much clearer and improved in all aspects.

This manuscript is a resubmission of an earlier submission. The following is a list of the peer review reports and author responses from that submission.

Round 1

Reviewer 1 Report

Comments and Suggestions for Authors

The manuscript presented by the authors addresses an interesting topic. They use massive sequencing techniques to try to answer questions of interest such as the usefulness of viaducts in maintaining connectivity between populations.

However, I think that the way the authors present their results is not sufficiently clear to me and therefore I am hesitant to accept the work as it is.

I am missing in the introduction some more information on the biology and the genetics of the species the authors worked with in this article. How far away they can travel and how genetically different are the species in the two areas analyzed? (If there's no previous information on that it would be nice that the authors pointed something about it, because the way in which data can be treated will depend a lot on that. The difference issue is just mentioned for M. whiteheadi and not for the rest of the species. It would be nice to see some values of genetic difference between areas such as Fst values or similar. If the two areas are genetically different and we have just control samples from one area and viaduct samples from the other the differences we can detect in genetics can be due to the area of origin and not to the "treatment". Also within and between "treatments" some sampling points look very close and others very far away. How that affects to the genetics of the different species?

I have problems with the colors used in the graphs, for example in figures 1,2 and 3 the red are the sampling points without viaduct but in figure 5 it represents the control sampling points. It is confusing

I know that sometimes it is a problem of size but, in figure 1 are really C1, C3 and C6 500m apart from the road?

My main concern is about table S1 sites names there doesn't match with the names in figure 1 and therefore is very difficult to see which samples belong to which "treatment". However I detect in the table S1 the lack of individuals from some areas that don't match with the 3 types of "treatment" in figure 5d. For example, for Tupaia glia there's no samples collected in any C location for Kenyir nor samples collected in I sites for Sungai Yu, so I would expect that figure 5d had only two treatments for any of the two locations instead of three. Same happens in other cases. But it is difficult to be sure since names, as I said before, doesn't match with figure 1. 

I also have concerns about the two-way ANOVA. Two many groups and very few individuals (or even none) in some groups can be a problem. More information about the results on the ANOVA would be needed.

Author Response

Reviewer #1

The manuscript presented by the authors addresses an interesting topic. They use massive sequencing techniques to try to answer questions of interest such as the usefulness of viaducts in maintaining connectivity between populations.

>> We thank Reviewer #1 for his appreciation of the interest and usefulness of our manuscript.

However, I think that the way the authors present their results is not sufficiently clear to me and therefore I am hesitant to accept the work as it is.

I am missing in the introduction some more information on the biology and the genetics of the species the authors worked with in this article. How far away they can travel and how genetically different are the species in the two areas analyzed? (If there's no previous information on that it would be nice that the authors pointed something about it, because the way in which data can be treated will depend a lot on that.

>> We thank Reviewer #1 for his suggestion in providing biological and genetic details about our studied species. However, we failed to extract useful information from previous studies to highlight their dispersal patterns at the study areas. In the revised manuscript, we calculated genetic differentiation (FST) and relatedness of individuals between the two study areas and found that there is no significant differentiation for all four species (Lines 264-269).

The difference issue is just mentioned for M. whiteheadi and not for the rest of the species. It would be nice to see some values of genetic difference between areas such as Fst values or similar. If the two areas are genetically different and we have just control samples from one area and viaduct samples from the other, the differences we can detect in genetics can be due to the area of origin and not to the "treatment". Also within and between "treatments" some sampling points look very close and others very far away. How that affects to the genetics of the different species?

>> We understand Reviewer #1’s concern here. As mentioned in the response to the previous comment, we calculated FST values to find low differentiation between two study areas across all four species. Results show all four species exhibit low genetic differentiation between the two study areas (MR = 0.055480442; MW = 0.022113378, NC = 0.069197229, TG = 0.015547675). Moreover, we found related individuals (relatedness > 0.1) between the two study areas in MR and TG, suggesting occasional exchange of individuals between the two study areas. This is common as the two areas are relatively near and genetic differentiation is not only the result of isolation by distance. That is why we chose to use individual-based spatial analyses to investigate the subtle differentiation (Lines 264-269).

I have problems with the colours used in the graphs, for example in figures 1,2 and 3 the red are the sampling points without viaduct but in figure 5 it represents the control sampling points. It is confusing.

>> We have changed the colour scheme so that viaduct sites are consistently blue, non-viaduct sites are red and control sites are green.

I know that sometimes it is a problem of size but, in figure 1 are really C1, C3 and C6 500m apart from the road?

>> Yes, all control sites are at least 500m from the road. The reason they look closer to the road is because we had to make the points bigger so that they would be visible on the map. We have moved the points farther from the road for clarity.

My main concern is about table S1 sites names there doesn't match with the names in figure 1 and therefore is very difficult to see which samples belong to which "treatment". However I detect in the table S1 the lack of individuals from some areas that don't match with the 3 types of "treatment" in figure 5d. For example, for Tupaia glia there's no samples collected in any C location for Kenyir nor samples collected in I sites for Sungai Yu, so I would expect that figure 5d had only two treatments for any of the two locations instead of three. Same happens in other cases. But it is difficult to be sure since names, as I said before, doesn't match with figure 1. 

>> The grid names have an extra letter at the end to denote whether they were on the north or south side, or east or west side of the road. I have removed this letter to reflect the site names (pair of grids).

I also have concerns about the two-way ANOVA. Too many groups and very few individuals (or even none) in some groups can be a problem. More information about the results on the ANOVA would be needed.

>> We agree with Reviewer #1 that low sample size may affect the accuracy of ANOVA. In the revised manuscript, we have replaced our ANOVA with Kruskal-Wallis tests. As there are only small numbers of individuals and none in some sites and unbalanced data between the sites, we think there is unlikely enough data to fit ANOVA models, especially as two-way models. We have also indicated the sample sizes for each comparison (Lines 232-235, 281-298).

Reviewer 2 Report

Comments and Suggestions for Authors

Dear Authors

I carefully read your manuscript "Landscape genomics highlights viaducts as potential dispersal conduits for fragmented small mammal populations". The research reported in the manuscript is very interesting, and has a high potential for applicability in conservation biology. I will recommend its publication, and I have a few recommendations for improving the manuscript.

>>> Add more keywords. They help readers understand the content of the paper. Furthermore, keywords make the article more visible to search engines such as Google Scholar.

>>> Figure 1 needs to be modified. Add to the map: 1) The study area in relation to its continent; 2) the area of study in relation to the world. The Figure does not clearly indicate where the research was carried out.

>>> Finally, please make abundant use of any graphic elements - photos, maps, infographics, drawings to broaden readers' understanding of the several important information and messages in this chapter. Willian Zinsser, in the excellent How to Write Well: The Classic American Handbook of Journalistic and Nonfiction Writing, says that the writer's task is to make life easier for readers - so images are very welcome in this paper. Obvious images are those of individuals of the species Maxomys rajah, Maxomys whiteheadi, Tupaia glis and Niviventer cremoriventer. Why are there no photos of those animals in the manuscript? It is true that illustrations of those animals appear in figure 3, but they are tiny and barely legible.

Cheers,

The Reviewer

Author Response

Reviewer #2

Dear Authors

I carefully read your manuscript "Landscape genomics highlights viaducts as potential dispersal conduits for fragmented small mammal populations". The research reported in the manuscript is very interesting, and has a high potential for applicability in conservation biology. I will recommend its publication, and I have a few recommendations for improving the manuscript.

>>> Add more keywords. They help readers understand the content of the paper. Furthermore, keywords make the article more visible to search engines such as Google Scholar.

>> We thank Reviewer #2 for the suggestion in our keywords. We have added “conservation planning”, “protected area”, “local population extinction”.

>>> Figure 1 needs to be modified. Add to the map: 1) The study area in relation to its continent; 2) the area of study in relation to the world. The Figure does not clearly indicate where the research was carried out.

>> We agree with Reviewer #2. In the revised manuscript, we added an inset map of Asia in Figure 1c, with Peninsula Malaysia highlighted in red.

>>> Finally, please make abundant use of any graphic elements - photos, maps, infographics, drawings to broaden readers' understanding of the several important information and messages in this chapter. Willian Zinsser, in the excellent How to Write Well: The Classic American Handbook of Journalistic and Nonfiction Writing, says that the writer's task is to make life easier for readers - so images are very welcome in this paper. Obvious images are those of individuals of the species Maxomys rajah, Maxomys whiteheadi, Tupaia glis and Niviventer cremoriventer. Why are there no photos of those animals in the manuscript? It is true that illustrations of those animals appear in figure 3, but they are tiny and barely legible.

>> We thank Reviewer #2 for the suggestion on visual improvement for the manuscript. We added a new figure (Figure 3 in the revised manuscript) with clear photos of our target species.

Round 2

Reviewer 1 Report

Comments and Suggestions for Authors

In this new version of the manuscript, the authors have addressed and answered most of the comments made by the reviewers.  But I think that the manuscript still have important flaws that must be corrected by the authors before being accepted and therefore I recommend rejecting it in its present form.

One of my main concerns remains unanswered. It is impossible to know where each individual has been sampled beyond the area of origin. If we take into account that, as the authors point out, we are talking about individual-based spatial analysis I think it is essential to know where each individual has been collected. That is impossible if the sampling sites in the supplementary table S1 and in Figure 1 do not match. Where is located the F1 site from the supplementary table in Figure 1? Is it V1, C1 or NV1? You should change the F, C and I sites from the table to NV, V and C as in figure 1. And I also believe that the letter at the end indicating whether they were on the north or south side, or on the east or west side of the road should be kept.

Regarding the new figures 6 and 7 it was not clear enough to me what genetic distance comparisons are used for the graphs. For example, in Figure 7b for the Kenyir population there is an n=10 and the individuals are 7 (2 in F1S, 1 in F1N, 3 in I2S and 1 in I2N) Are you using the distances of all pairs where one individual is S and the other N? but then shouldn't the number of comparisons for Sungay Yu be greater than 80? To study the role of the study areas (figure 7) ¿would it not be better to use all the comparisons?

In Figure 6, do all treatments for all species include individuals from both study areas? If not, how can we be sure that we are not including the effect of areas differences in the treatment?

Also I am not as confident as the authors in claiming in Figure 5 that there is a clear separation between the two populations when there is even overlapping of individuals in the central area of the graph were most of the individuals from both populations are located. I would tend to be more cautious with that statement.